# Chemical Constituents, Antioxidant Potential, and Antimicrobial Efficacy of *Pimpinella anisum* Extracts against Multidrug-Resistant Bacteria

**DOI:** 10.3390/microorganisms11041024

**Published:** 2023-04-14

**Authors:** Aisha Nawaf AlBalawi, Alaa Elmetwalli, Dina M. Baraka, Hadeer A. Alnagar, Eman Saad Alamri, Mervat G. Hassan

**Affiliations:** 1Biology Department, University College of Haqel, University of Tabuk, Tabuk 71491, Saudi Arabia; 2Department of Clinical Trial Research Unit and Drug Discovery, Egyptian Liver Research Institute and Hospital (ELRIAH), Mansoura 35818, Egypt; 3Botany and Microbiology Department, Faculty of Science, Benha University, Benha 33516, Egypt; 4Nutrition and Food Science Department, University of Tabuk, Tabuk 71491, Saudi Arabia

**Keywords:** MDR bacteria, bioactive compound, phenolics, *Pimpinella anisum*, GC–MS, antioxidant, *trans*-anethole, estragole

## Abstract

Aniseeds (*Pimpinella anisum*) have gained increasing attention for their nutritional and health benefits. Aniseed extracts are known to contain a range of compounds, including flavonoids, terpenes, and essential oils. These compounds have antimicrobial properties, meaning they can help inhibit the growth of nasty bacteria and other microbes. The purpose of this study was to determine if aniseed extracts have potential antioxidant, phytochemical, and antimicrobial properties against multidrug-resistant (MDR) bacteria. A disc diffusion test was conducted in vitro to test the aniseed methanolic extract’s antibacterial activity. The MIC, MBC, and inhibition zone diameters measure the minimum inhibitory concentration, minimum bactericidal concentration, and size of the zone developed when the extract is placed on a bacterial culture, respectively. HPLC and GC/MS are analytical techniques used for identifying the phenolics and chemical constituents in the extract. DPPH, ABTS, and iron-reducing power assays were performed to evaluate the total antioxidant capacity of the extract. Using HPLC, oxygenated monoterpenes represented the majority of the aniseed content, mainly estragole, *cis*-anethole, and *trans*-anethole at 4422.39, 3150.11, and 2312.11 (g/g), respectively. All of the examined bacteria are very sensitive to aniseed’s antibacterial effects. It is thought that aniseed’s antibacterial activity could be attributed to the presence of phenolic compounds which include catechins, methyl gallates, caffeic acid, and syringic acids. According to the GC analysis, several flavonoids were detected, including catechin, isochiapin, and *trans*-ferulic acid, as well as quercitin rhamnose, kaempferol-*O*-rutinoside, gibberellic acid, and hexadecadienoic acid. Upon quantification of the most abundant estragole, we found that estragole recovery was sufficient for proving its antimicrobial activity against MDR bacteria. Utilizing three methods, the extract demonstrated strong antioxidant activity. Aniseed extract clearly inhibited MDR bacterial isolates, indicating its potential use as an anti-virulence strategy. It is assumed that polyphenolic acids and flavonoids are responsible for this activity. *Trans*-anethole and estragole were aniseed chemotypes. Aniseed extracts showed higher antioxidant activity than vitamin C. Future investigations into the compatibility and synergism of aniseed phenolic compounds with commercial antibacterial treatments may also show them to be promising options.

## 1. Introduction

In recent years, multidrug resistance has emerged as a major challenge not only in chemotherapy, but also in antibiotic therapy due to the emergence and spread of bacterial pathogens [1]. Nevertheless, in addition to intrinsic antibacterial agents [2], certain medicinal plants also generate inhibitors of multidrug resistance [3], which are capable of enhancing the activity of antibiotics against multidrug-resistant bacteria [4]. As a result of this finding, crude extracts were examined for potential synergistic interactions with common antibiotics against resistant bacteria, opening the door for the discovery of plant-based multidrug resistance inhibitors [5].

Medicinal plants remain a reliable source of compounds with medicinal properties despite these problems [6]. In addition to being widely studied, these plants have a considerable economic interest because they are widely used in agriculture, pharmaceuticals, and cosmetics, as they contain a wide variety of antimicrobial and antioxidant compounds [7]. Infectious diseases and microbial pathogenicity have been treated with natural products for thousands of years worldwide, prior to the introduction of antibiotics and other modern medications [8]. Specifically, aniseed (*Pimpinella anisum*) is used to treat osteoarthritis, gastritis, skin irritation, and dental aches [9]. Aside from being anti-flu and anti-HIV, it has also been reported to possess antibacterial, antiseptic, insecticidal, and chemopreventive properties [10]. Catechin, isochiapin, *trans*-anethole, estragole, and trans-ferulic acid, as well as quercitin rhamnoside, are commonly occurring chemicals that have been previously discovered in aniseed extract [11].

Secondary metabolites of plants are believed to play a role in the plant’s defense system against pathogens, as they are capable of inhibiting the growth of microorganisms and reducing the risk of infection. Some active compounds in medicinal plants, such as alkaloids, terpenoids, and flavonoids, have been found to have anti-inflammatory, anti-tumor, antimicrobial, and analgesic properties [12,13]. Furthermore, the compounds in plants are natural and have fewer side effects than synthetic drugs, making them a safer and more sustainable way to treat diseases [14,15]. Plants have been used therapeutically in Egyptian culture for thousands of years. The active compounds in medicinal plants have been studied to understand their pharmacological effects, and they have been found to have antibacterial, antifungal, and antiviral properties [16,17]. This makes them a safe and effective way to treat infections. Additionally, they are generally more affordable than synthetic medications, making them more accessible to people who may not otherwise have access to treatment [18,19].

Thus, this in vitro study was conducted to determine the effectiveness of aniseed methanolic extract against multidrug-resistant bacteria as well as to determine the antioxidant potential of the extract and its phytochemical composition by using comprehensive GC–MS and HPLC methods to identify the extract’s bioactive constituents and phenolic compounds. 

## 2. Materials and Methods

### 2.1. Preparation of Plant Materials Extracts

The aniseed plants were collected from the Horticulture Research Institute, Agriculture Research Center, Giza, Egypt. Plant parts were gently washed with distilled water and air-dried for three days at room temperature before being carefully ground in a blender. An amount of 10 g of ground, air-dried plant material was soaked in 50 mL of methanol in conical flasks, and then incubated at room temperature for 72 h with shaking at 120 rpm. Centrifugation of the crude extracts was carried out at 3354× *g* for 10 min at 25 °C then evaporated at 80 °C in a rotary evaporator. Then, the extracts were stored at 4 °C for further experiments [20].

### 2.2. Antibiotic Susceptibility and Bacterial Strains

A number of 37 bacterial samples were taken from patients presenting at the Benha University Hospital in Benha, Egypt. These 37 bacteria from gram-positive and gram-negative groups were taken from a variety of clinical specimens, including urine, prostatic secretion, and wound discharge. We maintained all bacterial isolates on nutrient agar slants except *Enterococcus faecalis*, which was cultured on trypticase soya agar (TSA). Isolates were regularly subcultured and stored in 10% glycerol suspension at 4 °C and at −80 °C to ensure recovery when needed.

Then, adopting several antibiotics that perceive cell walls, protein synthesis, and DNA, all clinical isolates were tested for antibiotic resistance using the Kirby–Bauer disc diffusion method [21]. Penicillin, amoxicillin/clavulanate, vancomycin, ampicillin/sulbactam, nitrofurantoin, aztreonam, cefoperazone, chloramphenicol, clindamycin, gentamicin, tetracycline, erythromycin, ofloxacin, norfloxacin, and trimethoprim/sulfamethoxazole were the antibiotics used to test the sensitivity of the isolated strains. At least one resistant strain from each of the examined organisms that was resistant to the aforementioned antibiotics was chosen for further evaluation and biochemically identified. Inhibition zones (mm) were categorized as sensitive, intermediate, or resistant.

### 2.3. Identification of Bacterial Isolates by VITEK2 System

Suspensions were prepared by emulsifying bacterial isolates in 0.45% saline to the equivalent of a 0.5 McFarland turbidity standard. VITEK^®^2 Systems 7.01 software (BIOMÉRIEUX) was used to confirm the conventional biochemical identification of the selected bacterial isolates [22]. Substrates presented in the identification card specific for gram-negative identification used in the VITEK^®^2 system were as follows: APPA, ala-phe-pro-arylamidase; ADO, adonitol; PyrA, l-pyrrolydonyl-arilamidase; IARL, l-arabitol; dCEL, d-cellobiose; BGAL, beta-galactosidase; H_2_S, H_2_S production; BNAG, beta-n-acetyl-glucosaminidase; AGLTp, glutamylarylamidase PNA; dGLU, d-glucose; GGT, gamma-glutamyl-transferase; OFF, fermentation glucose; BGLU, beta-glucosidase; dMAL, d-maltose; dMAN, d-mannitol; dMNE, d-mannose; BXYL, beta-xylosidase; BaLAP, beta-alaninearylamidase pna; ProA, l-proline arylamidase; LIP, lipase; PLE, palatinose; TyrA, tyrosine arylamidase; URE, urease; dSOR, d-sorbitol; SAC, saccharose/sucralose; dTAG, d-tagatose; DTRE, d-trehalose; CIT, citrate (sodium); MNT, malonate; 5 KG, 5-keto-d-gluconate; ILATk, L-lactate alkalinization; AGLU, alpha-glucosidase; SUCT, succinate alkalinization; NAGA, beta-n-acetyl-galactosaminidase; AGAL, alpha-galactosidase; PHOS, phosphatase; GlyA, glycine arylamidase; ODC, ornithine decarboxylase; LDC, lysine decarboxylase; ODEC, DECARBOXYLASE base; IHISa, L-histidine assimilation; CMT, coumarate; BGUR, beta-glucoronidase; O129R, o/129 resistance; GGAA, glu-gly-arg-arylamidase; IMLTa, L-MALATE assimilation; ELLM, ellman; and ILATa, L-lactate assimilation.

### 2.4. Evaluation of Antimicrobial Susceptibility Testing, Minimum Inhibitory Concentrations (MIC), and Minimum Bactericidal Concentrations (MBC)

In order to determine whether or not the plant extracts have antibacterial properties, the disc diffusion procedure for assessing antimicrobial susceptibility was used. This testing was carried out in accordance with the standard method as described previously [23]. Plant extracts were tested for their ability to inhibit the growth of the bacteria with the greatest level of resistance to several drugs [23]. MIC values were determined for plant extracts with antibacterial activity according to the microplate method of Eloff (1998) [24]. In this study, a 25 mg/mL concentration of the plant extract was prepared and dissolved in a 10% solution of dimethyl sulfoxide (DMSO). A 96-well polystyrene flat-bottomed multi-well microplate (Sigma Aldrich, St. Louis, MO, USA) was designed to test each extract at 1000 µg/mL and serially dilute twice to 15.6 µg/mL. Each well was then filled with 10 µL (1 × 10^6^ CFU/mL) of bacteria. An ELISA reader was used to read pre-incubation absorbance values. In each experiment, the antibiotic amikacin was utilized as a reference antibiotic, while an extract-free solution served as the negative control. After incubation at 37 °C for 24 h, the microplates were read for absorbance and MIC values and the experiment was repeated twice. By streaking on the surface of solid nutrient agar, bacterial cells were transferred from the MIC plate and subcultured. After a 24 h period of overnight incubation at 37 °C, MBC was calculated [25].

### 2.5. Time-Kill Kinetics

Following the method described in the previous work by Akinjogunla et al. [26], a start-up inoculum containing between 10^5^ and 10^6^ CFU/mL was obtained by subculturing and diluting the selected bacteria to 0.5 McFarland standard turbidity. For 90 min, the tubes were shaken continuously at 150 rpm at 37 °C to ensure that microbial growth was in the logarithmic phase (exponential). To initiate growth in test tubes, concentrations equal to the MICs of aniseed extract were prepared and transferred into sterile broth. This was followed by an inoculum (density ∼10^5^–10^6^ CFU/mL), and then incubation at 37 °C. CFU/mL was calculated by taking aliquots after (0, 0.5, 1, 1.5, 2, 3, 4, and 6 h).

### 2.6. HPLC Analysis of Phenolic Contents

As part of the investigation, standard solutions were formulated for investigating phenolic compounds: estragole, anethole, *trans*-anethole, catechin, methyl gallate, caffeic acid, syringic acid, kaempferol-*O*-rutinoside, rosmarinic acid, quercetin, *O*-coumaric acid, naringenin, ferulic, syringic, vanillic, caffeic, ellagic, and gallic acids (Sigma-Aldrich, Merck, Darmstadt, Germany). Using the HPLC grade methanol (Ultragradient grade; Sigma-Aldrich, Merck, Darmstadt, Germany) as a solvent for dissolving the standards, stock solutions of 100 mg/L were prepared, which were then used to produce 50, 40, 30, 20, and 10 mg/L working solutions. To determine the HPLC wavelength for each phenolic compound sample, we assessed their absorbance via a UV–Vis spectrophotometer (UV-2600; Shimadzu, Tokyo, Japan).

HPLC analysis was performed with the Agilent 1260 series armed with a UV–Vis. An Eclipse C18 reverse-phase chromatography column with a size of (5 µm, 250 × 4.6 mm) was used for separation. Mobile phases included water (A) and acetonitrile (B), which contained 0.05% trifluoroacetic acid. The mobile phase flow rate was 1 mL/min with a linear gradient as follows: starting with 95% A and 5% B; 5–8 min 65% A and 35% B; 8–12 min 65% A and 35% B; 12–15 min 50% A and 50% B; 15–16 min 30% A and 70% B; 16–20 min 95% A and 5% B. The multi-wavelength detector was monitored at 280 nm. The injection volume was 5 µL per sample solution and the column temperature was 40 °C [27].

### 2.7. Chemical Constituents Gas Chromatography–Mass Spectroscopy (GC–MS) Analysis

GC–MS was used to analyze the extract. In our protocol, trimethylsilylation was carried out by adding 100 µL of derivatization reagent (80 µL BFSTA + 20 µL TMCS) and incubating at 65 °C for 1 h. An HP-5 MS capillary column (30 m × 0.25 mm i.d., 0.25 µm film thickness) and a flame ionization detector were used for the GC analysis. As a carrier gas, helium flows at a rate of 1 mL/min. A temperature of 250 °C was used for the injector and a temperature of 280 °C for the detector. Initially, the column temperature was kept at 40 °C for 5 min. It was gradually raised to 250 °C at 2 °C/min for 5 min, then increased to 275 °C at 5 °C/min for 5 min. Finally, sample was injected into the system in split mode with a ratio of 10:1. According to [28], extract components were identified by the Wiley NIST 2011 mass spectral library of the GC–MS data system. 

### 2.8. Quantitative Analysis

Having performed qualitative analysis with GC–MS, the most abundant chemical constituent was then quantified using LC–MS/MS (6545, Agilent Technologies Inc., Santa Clara, CA, USA) equipped with an Agilent Eclipse plus C18 column (50 × 3.0 mm^2^, 1.8 µm) to ensure that the compound had the greatest effect on MDR bacteria.

#### 2.8.1. Preparation of Stock Solutions, Calibration Standards, and Quality Control Samples

In an amber-colored glass bottle at 4 °C, estragole (4-*allyl* anisole) was diluted in *n*-hexane to 50 mg/mL. An amber-colored glass bottle was used for storage of the internal standard (IS) stock solution of *p*-anisaldehyde at 1.0 mg/mL in *n*-hexane. Suitable dilutions of stock solution have been used to prepare calibration standards (0.20–20.00 ng/mL). *n*-hexane was used to prepare quality control samples (QC) of stock solutions at concentrations of 6, 100, and 280 ng/mL.

#### 2.8.2. Method Validation

ICH guidance was followed for the validation of the proposed analytical methodology for estragole quantitative analysis. An x-axis plot with estragole concentration and its peak area ratio on a y-axis was used to estimate estragole’s linearity range. A six-replication study was conducted to determine estragole linearity in the range of 0.20–20.00 ng/mL. According to the standard addition method, the accuracy of the estragole method was estimated as percent of recovery (% recovery). Each estragole concentration was estimated based on the % recovery. The method precision, accuracy, selectivity, and other parameters of validation were also assessed.

#### 2.8.3. Quantitative Analysis of Estragole Using LC–MS/MS Method

We used ammonium formate in 5 mM concentration (A) and a mixture of methanol and water (B) at a flow rate of 0.3 mL/min as the solvent composition. Starting at 95% A, a gradual decrease to 35% A took 13 min, then a gradual decrease to 0% A took 3 min, held for 4 min, then increased to a gradual return to 95% A after 2 min. At 40 °C of column temperature, 5 µL of aliquot were injected into the UPLC system. Desolvation gas flow rate—10 L/min, nebulizer gas pressure—40 psi, capillary voltage—4300 V, desolvation gas temperature—360 °C—were used in the MS detection condition. The MS analysis was performed using both electrospray ionization modes to obtain full scan mass spectra (*m*/*z* 50–750). The UPLC system was set to 40 °C in order to achieve optimal resolution and peak shape of the compounds being analyzed. The MS detection settings were carefully optimized in order to maximize sensitivity and selectivity of the MS analysis.

### 2.9. Assessment of Antioxidant Activity

#### 2.9.1. DPPH Scavenging Activity

A solution of 0.25 mL of DPPH methanol was mixed with 1 mL of aniseed seed extracts and allowed to stand for 30 min in the dark after shaking vigorously for 1 min. Each sample was incubated at 37 °C for 30 min. A UV spectrophotometer was used to measure absorbance at 517 nm [29]. The following equation was used to compute the percentage of DPPH radical scavenging activity:% DPPH radical scavenging activity = [(A_0_ − A_1_)/A_0_] × 100
where A_0_ represents the absorbance of the control and A_1_ represents the absorbance of the extract or standard.

#### 2.9.2. ABTS Radical Cation Scavenging Activity

ABTS radical cation scavenging activity was measured at different concentrations, and the results were compared with that of standard materials; ascorbic acid was measured at similar concentrations as discussed in the previous study by [30]. Based on the absorbance at 734 nm of ABTS radical cation scavenging activity, the following equation was calculated:% ABTS radical cation scavenging activity = [1 − (A_sample_/A_control_)] × 100

#### 2.9.3. Reducing Power Scavenging Assay

2.5 mL of sodium phosphate buffer (0.32 M, pH 6.6) and 2.5 mL of potassium ferricyanide were added to 1 mL of various concentrations of diluted plant extracts in methanol. After 20 min at 50 °C, the liquid was centrifuged for 10 min before 2.5 mL of 10% trichloroacetic acid was added. Using a mixture of 2.5 mL deionized water and 0.5 mL ferric chloride (0.1%), we measured the solution’s absorbance at 700 nm in comparison to a blank [31].

### 2.10. Statistical Analyses

In all experiments, the results were expressed as mean ± (SD). To compare means, a one-way analysis of variance (ANOVA) was conducted followed by a post hoc Tukey test at a significance level of 5%.

## 3. Results

### 3.1. Antibiotic Susceptibility and Bacterial Strains

Based on the test results of 32 gram-positive and gram-negative bacteria against 16 antibiotics, (Table 1) shows that isolates exhibited antibiotic resistance and there was a high and alarming level of antibiotic resistance among the tested isolates in this study. The most common antibiotics showing resistance were penicillin, norfloxacin, clindamycin, and trimethoprim/sulfamethoxazole. This suggests that due to the improper and overuse of antibiotics, bacteria are developing resistance to these antibiotics, making them less effective at treating infections. This could be a major public health concern as it could lead to more serious illnesses and even death. 

### 3.2. Identification of Bacterial Isolates by VITEK2 System

Further identifications were carried out by the VITEK2 system, the biochemical identification is summarized in (Appendix A); the results of susceptibility testing (Table 2, Table 3, Table 4, Table 5 and Table 6) for bacterial isolates by the VITEK 2 method were compared with those of the manual method using pure cultures according to CLSI. The data revealed that there was a high similarity between the manual susceptibility method agreement and the VITEK 2 system for the antibiotics used.

### 3.3. Antimicrobial Susceptibility Testing for the Methanolic Extract of Aniseed

Methanolic extracts of aniseed were tested using the agar disc diffusion method against five isolates that were highly drug-resistant (Table 7). Compared to amikacin as the gold standard test, methanolic extracts of aniseed significantly inhibited all of the test bacteria. Statistical differences in inhibition zones were found for all bacterial tests at experimental concentrations of extracts (Table 7). The inhibitory zone of *Pseudomonas aeruginosa* was the largest (22.86 mm), indicating that phenolic compounds may have a greater impact on its antibacterial activity. However, *Acinetobacter baumannii* (17.56 mm) was considered to be the least sensitive. As for the MICs and MBCs, the MIC values varied with the bacterial species. However, in general, the values ranged from 0.095 to 0.170 mg/mL, indicating that the extracts had high potency. There was a trend of inhibition against all bacteria in aniseed extracts (0095–0.130 mg/mL). MBC assays confirmed the results of disc diffusion assays as well as MIC determinations. Aniseed extracts were found to have bactericidal activity as indicated by their MBC values of 0.190–0.260 mg/mL as revealed in (Table 8). The selected isolates (Gram-negative: *Salmonella typhi*, *Pseudomonas aeruginosa*, and *Acinetobacter baumannii*) and (Gram-positive: *Enterococcus faecalis* and *Staphylococcus aureus*) were sensitive to aniseed extract based on the MIC and MBC assays.

### 3.4. Time-Kill Kinetics

Aniseed methanolic extracts were tested for killing time against each bacterial isolate, and results can be found in Figure 1A–E. It was found that aniseed extracts had a greater effect on *Enterococcus faecalis*, *Staphylococcus aureus*, *Salmonella typhi*, *Pseudomonas aeruginosa*, and *Acinetobacter baumannii*, all of which were killed within 3 h. *Pseudomonas aeruginosa*, however, was killed approximately one hour after the aniseed extract was added.

### 3.5. HPLC Analysis of Phenolic Contents

In the study, polyphenols extracted from aniseeds were characterized qualitatively and quantitatively by HPLC. Polyphenols were identified by comparing the retention times of peaks with those of standard compounds. By comparing the peak areas of the identified compounds with standards, the quantification of the compounds was performed. A number of 25 phenolic compounds have been determined by HPLC from aniseed extracts. In fact, oxygenated monoterpenes represent the vast majority of the aniseed content, mainly estragole, *cis*-anethole, and *trans*-anethole with concentrations of 4422.39, 3150.11, 2312.11 (µg/g), respectively, as revealed in (Table 9) and (Figure 2).

### 3.6. GC Analysis

In this study, gas chromatography was used to analyze aniseed’s chemical composition. GC–MS analysis of aniseed revealed more than 27 isolated components. The retention times and peak area (%) of the main isolated components in (Table 10) are shown for the main isolated component estragole, which represents more than 65% of the isolated components (Figure 3).

### 3.7. Method Validation Outcomes

In accordance with ICH recommendations, a number of validation parameters were examined for estragole quantitative analysis. In Table 11, we present the results of the linear regression analysis of estragole. The results showed an excellent linear fit, with an R-squared value of 0.9992 for the range of 0.20–20.00 ng/mL, indicating that estragole levels can be accurately determined with the developed method.

**Table 11 microorganisms-11-01024-t011:** Characteristics outcome of the validated method carried.

Technique	LC–MS/MS
Selectivity	No interferences from endogenous plasma samples (Figure 4).
Lower Quantification Limit	0.2 ng/mL.
Linearity and Calibration Range	R = 0.9992 for range 0.20–20.00 ng/mL.
Stability%:	
a. Short-Term (6 h)	Between 93.66% and 97.10%.
b. Stock Solution	Mean 91.03% and 109.86% for internal standard and analyte, respectively, left for 5 h at RT. Mean 98.29% and 101.73% for internal standard and analyte, respectively, left for 27 days and in the refrigerator.
c. Auto-Sampler (9.5 h)	Between 94.87% and 98.77% for analyte and 105.38% for internal standard.
d. Dry Extract	Between 100.53% and 101.09% for analyte and 98.96% for internal standard (IS) left dry 1.00 h at RT. Between 97.81% and 99.31% for analyte and 101.43% for IS for 50.0 h in the refrigerator.
Recovery	
a. Absolute Recovery	Between 61.78% and 72.36% and 83.88%.
b. Relative Recovery	Between 101.78% and 104.66%.
Intra-Batch Precision (% CV)	Between 2.39% and 4.79% and 3.92% for LLOQ.
Intra-Batch Accuracy%	Between 96.22% and 103.10% and 89.33% for LLOQ.
Inter-Batch Precision (% CV)	Between 4.14% and 5.54% and 13.19% for LLOQ.
Inter-Batch Accuracy%	Between 100.00% and 104.75% and 92.33% for LLOQ.
Dilution Integrity	Accuracy between 92.60% and 101.17%.

### 3.8. Quantitative Analysis of Estragole

Estragole was quantified by LC–MS/MS based on the standard concentration. The standard concentration was used to create a calibration curve, which was then used to determine the concentration of estragole in the sample (Figure 5). Table 12 shows the results of this analysis.

Table 12 shows the concentration and confirmation of estragole in the aniseed sample as determined by LC–MS/MS. Based on the mass spectra obtained, the molecular ion at m/z 150 matched those of a standard estragole. This illustrates how estragole was identified using LC–MS/MS by mass spectra (Figure 6A), retention time of estragole, and spiked IS extracts (Figure 6B,C).

### 3.9. Total Antioxidant Activities

Based on the DPPH test, aniseed was compared to ascorbic acid for its ability to scavenge free radicals. There were statistically significant differences between aniseed methanolic extract and vitamin C, as revealed in (Figure 7a). Surprisingly, the presented results showed that aniseed methanolic extract appeared promising at scavenging DPPH radicals. When aniseed extraction concentration was increased from 100 to 1000 g/mL, the scavenging ability was significantly improved. A further study showed 15.18 µg/mL of extract had an IC_50_ in DPPH. Furthermore, (Figure 7b) shows the ABTS radicals at different concentrations. Aniseed extract showed activity at the lowest concentration, then gradually increased in activity with increasing concentration, with statistical differences compared with vitamin C. Aniseed extract had an IC_50_ of 19.27 µg/mL in the ABTS. Additionally, aniseed extract was tested for Fe^3+^ reductive ability using the Fe^3+^-Fe^2+^ transformation and compared to ascorbic acid as a reference material (Figure 7c). As the concentration of aniseed extracts increased, the reducing power increased. However, all of the extract concentrations showed lower activities than the control, and these differences were not statistically significant.

## 4. Discussion

In this study, antibiotic resistance was found to be high and alarming in tested isolates. Among them, five isolates showed the highest resistance to multiple antibiotics. Further identifications were undertaken by the VITEK2 system; the results of susceptibility testing for bacterial isolates by the VITEK 2 [32] method were compared with those of the manual method using pure cultures according to CLSI [33]. There was a high similarity between the manual susceptibility method agreement and the VITEK 2 system for the antibiotics used.

Although all of the bacteria exhibited some level of resistance to antibiotics, it was believed that penicillin, aztreonam, clindamycin, and erythromycin were the drugs to which the majority of the isolates were resistant. It is possible that antibiotic-resistant bacteria are simply normal bacteria that have undergone mutation as a result of widespread usage of broad-spectrum antibiotics [34]. It is typical practice for medical facilities such as hospitals and general clinics to administer antibiotics prior to obtaining results of culture and sensitivity tests [35]. It is also essential to keep in mind that genetic testing requires the allocation of large amounts of both technical and financial resources, which many clinical labs often lack [36].

Since the organisms that cause the disease and the rate of antibiotic resistance change over time and place, recent data may help physicians decide on best treatments [37]. This may help patients to receive the proper antibiotics, while the overuse of antibiotics, which leads to fast development of resistance, will be kept to a minimum [38,39].

In the current study, five isolates of the most drug-resistant bacteria were tested for antibacterial activity using aniseed extracts. The results of the experiment indicated that aniseed extract had a powerful effect on the bacteria tested. For instance, it inhibited the growth of *Pseudomonas aeruginosa*, the most sensitive species investigated, by producing a statistically significant inhibition zone. This assumed that phenolic compounds have a role in the antimicrobial activity of the aniseed extract.

However, our data revealed that *Acinetobacter baumannii* (17.56 mm) exhibited lesser sensitivity to the aniseed extract. Similarly, a study conducted by Kovač et al. [40] revealed that *Staphylococcus aureus* was more resistant to the antibacterial effects of both phenolic extracts of *Alpinia katsumadai* seeds and post-distillation extracts against *Campylobacter jejuni* [40]. Therefore, further research is necessary to examine how phenolic extracts affect bacteria and how they contribute to antimicrobial activity based on these findings.

It has been claimed that aniseed extract has antifungal qualities because of its anethole concentration [41,42]. Additionally, aniseed fractions in diethyl ether showed substantial antibiotic activity against MRSA, *Pseudomonas aeruginosa*, and *Acinetobacter baumannii* [43,44]. Additionally, *anethole* had far greater activity than other key components including anisyl alcohol, anisyl aldehyde, and anisyl acetone [45]. The non-polar components of essential oils have received the most attention in research on aniseed (low molecular weight volatile phenolic compounds) [46]. Moreover, it has been shown that aniseed’s non-polar components produce antimicrobial metabolites. Terpenoids and isoprenoids, which make up the bulk of plant essential oils and are the most varied class of biogenic volatile organic chemicals in plants [47], are greatly influenced by environmental and biotic variables, including light, temperature, soil water, and fertility [44].

Our data revealed the intriguing result that methanolic extract of aniseed inhibited the five selected isolates, especially *Enterococcus faecalis* and *Pseudomonas aeruginosa*. The MIC values for the extract against *Enterococcus faecalis* and *Pseudomonas aeruginosa* were determined to be approx. 95 and 125 mg/L, respectively, whereas the MBC was 190 and 250 mg/L, respectively. The results demonstrated substantial inhibition with promising antibacterial characteristics. The outcomes showed that the extract has a bactericidal action against the isolates that were tested. Moreover, measurements of killing time by the aniseed methanolic extract were shown in our study. At approximately 120 min, a methanolic extract of aniseed killed all isolates; however, the high activity of aniseed extract was revealed in the isolate of *Pseudomonas aeruginosa*. These results could be explained by the fact that the accelerated bacterial efflux pumps may be inhibited by the extract and shorten the time needed for the aniseed extract to diffuse inside the bacteria, contributing to bacterial inhibition [48,49].

The level of phenolic compounds found in plant tissues and the antibacterial activity of plant extracts are directly correlated with the pathogen’s level of resistance [50]. According to our findings, the principal components of the aniseed extract were estragole (4422.39 µg/g), *cis*-anethole (3150.11 µg/g), *trans*-anethole (2312.11 µg/g), and caffeic acid (652.32 µg/g), as determined by HPLC. The aniseed extract by HPLC is higher in phenolic compound content, which may explain its comparatively higher antibacterial activity. This study provided evidence that aniseed extract contains active inhibitors, including phenolic compounds, which could explain why these extracts are antibacterial. A similar pattern of findings was revealed for various plant materials as reported by [51], who demonstrated that furfural and phenolic components (mostly benzenetriol) are correlated to the antibacterial activity of pomegranate peels, jackfruit peels, and custard apple peels.

In our study, the high activity of aniseed extracts against these isolates may be due to their phenolic structure, as the hydroxyl groups in their structures make it possible for them to pass into the cell and permeabilize the cytoplasmic membrane [52], leading to distressed cellular metabolism [53]. The quantification of estragole in our study showed that the concentrations of this compound were high enough to prove that it was capable of inhibiting the growth of multidrug-resistant (MDR) bacteria. This indicates that estragole may be an effective antimicrobial agent against these types of bacteria. These results prove that other studies revealed that the majority of trans-anethole, fenchone, estragole, and limonene was also observed in the composition of aniseed essential oils [54]. These findings showed that *trans*-anethole and estragole for aniseed extract were two separate chemotypes of aniseed essential oils in the current study. *Trans*-anethol and estragole are typically the two primary ingredients in aniseed [55]. The amount of these chemicals accumulated varied greatly according to the seed’s area and vegetative stage [56].

Our extract also contains the caffeic acid ester of 3-(3,4-dihydroxyphenyl)-lactic acid, which has shown possible antibacterial activity, especially with gram-positive bacteria such as MRSA [57]. The combination of rosmarinic acid and vancomycin may also be effective against MRSA [57]. Numerous epidemiological studies have demonstrated that flavonoids and their glycosides have versatile health benefits [58,59]. It is not surprising that flavonoids have broad-spectrum antibacterial activity both in vitro and in vivo because they are synthesised in response to microbial attacks in plants [60,61]. Instead of having a single unique site of action, there are numerous cellular targets; several flavonoids detected in our analysis, including catechin, apegenin-7-*O*-glucoside, rutin, quercitin rhamnoside, kaempferol-*O*-rutinoside, *trans-*ferulic acid, eucalyptol, gibberellic acid, isochiapin, hexadecadienoic acid, and luteolin, possess potent antibacterial activity. In addition to exerting antibacterial effects, eugenol rutinoside is a volatile phenolic compound in the phenylpropene class found in many plant species [62,63].

A previous study demonstrated that eugenol has antibacterial properties against bacteria resistant to antibiotics, such as *A. baumannii* and *Staphylococcus* spp. It has been shown that glycosylation of eugenol enhances its antibacterial properties, particularly against *Staphylococcus* spp. and *E. coli* [64].

In antioxidant studies, aniseed methanol extracts scavenged DPPH radicals better than vitamin C, with an IC50 of 17.92 g/mL. Similarly, [65] discovered that fennel extracts had potent antiradical properties. Bulgarian fennel seed extract had low antiradical activity (IC50 = 113.19 mL/L), according to [66]. Previous research has shown that essential oils and extracts of aniseed have antioxidant properties [46,67]. Our knowledge of these compounds from aniseed is limited. A metabolic process in the body creates an oxidative stress state that creates free radicals that can lead to body damage if they are at risky levels [68]. Due to its high phenolic and flavonoid content, aniseed extract showed promising activity as an antioxidant agent [69,70]. According to Danilenko et al. [71], aniseed polar fractions may be useful as antibiotics and antioxidants in biopharmaceuticals. As alternatives to the plant extracts, the natural metabolites of some lactobacilli possess antimicrobial, antifungal, antioxidant, and anti-inflammatory effects and can be used as a metabiotic in medicine.

In conclusion, the selected MDR bacterial isolates were significantly inhibited by aniseed extract; this suggests that it could be a promising natural compound for treating drug-resistant bacteria infections, providing it as a viable anti-virulence option. The mechanism of its antimicrobial action could be attributed to the presence of phenolic compounds in the extract. Further, this study revealed significant variation in the biochemical composition of Egyptian aniseed. Among the oxygenated monoterpenes of aniseeds analyzed by HPLC, estragole, *cis*-anethole, and *trans*-anethole constituted the majority of the aniseed extract. The quantification results of LC–MS/MS showed that estragole was present in higher concentrations. Furthermore, its presence in higher concentrations suggests that it could be the active compound responsible for the observed antibacterial effect. The antioxidant capacity of aniseed extracts was observed to be stronger than that of vitamin C, which suggests that it is a powerful antioxidant with potential health benefits. Additionally, the compounds in aniseed may be synergistic with existing antibacterial treatments, which could lead to more effective treatments for bacterial infections.

## Figures and Tables

**Figure 1 microorganisms-11-01024-f001:**
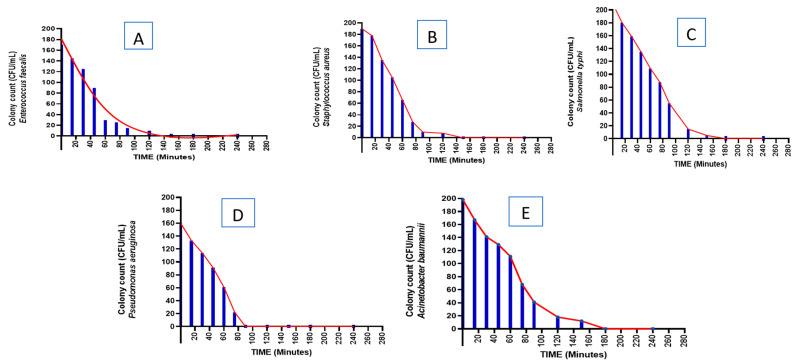
Time-kill experiment of (**A**) *Enterococcus faecalis*; (**B**) *Staphylococcus aureus;* (**C**) *Salmonella typhi*; (**D**) *Pseudomonas aeruginosa*; (**E**) *Acinetobacter baumannii.* After 3 h, the relative viable count of each isolate was measured against aniseed extract and expressed as CFU/mL (% of the control). All bacteria were in the logarithmic phase at the beginning of the experiment, and aliquots were taken at 0, 0.5, 1, 1.5, 2, 3, 4, and 6. The viable colony counts on blood agar were determined as CFU/mL.

**Figure 2 microorganisms-11-01024-f002:**
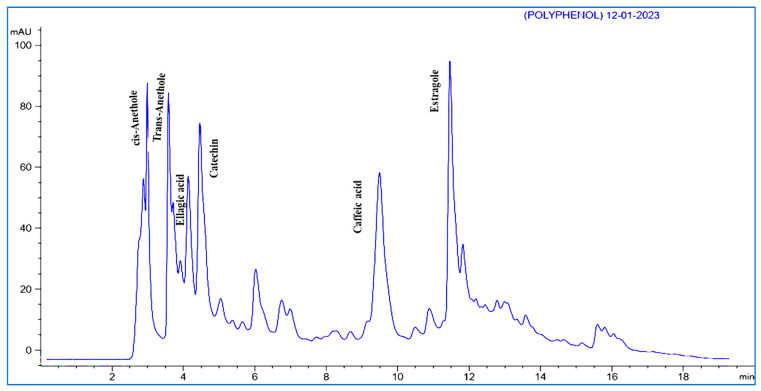
Chromatogram of aniseed HPLC. Estragole, cis-anethole, and trans-anethole had the highest RTs at 11.52, 2.82, and 3.55, respectively.

**Figure 3 microorganisms-11-01024-f003:**
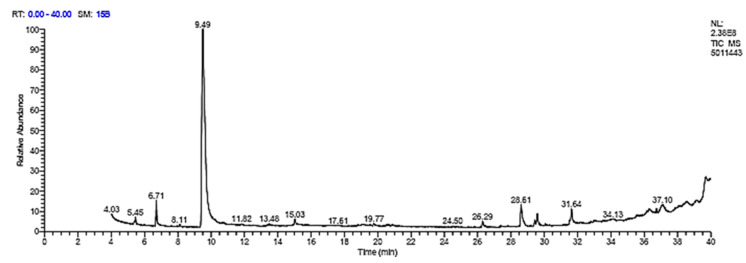
GC–MS spectrum of aniseed methanolic extract.

**Figure 4 microorganisms-11-01024-f004:**
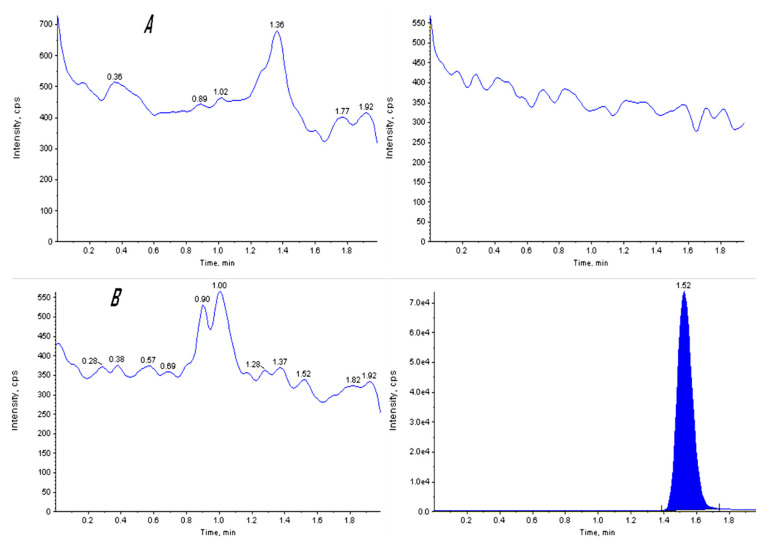
Representative chromatogram for selectivity of (**A**) blank standard and IS blank, and (**B**) zero standard and IS extract.

**Figure 5 microorganisms-11-01024-f005:**
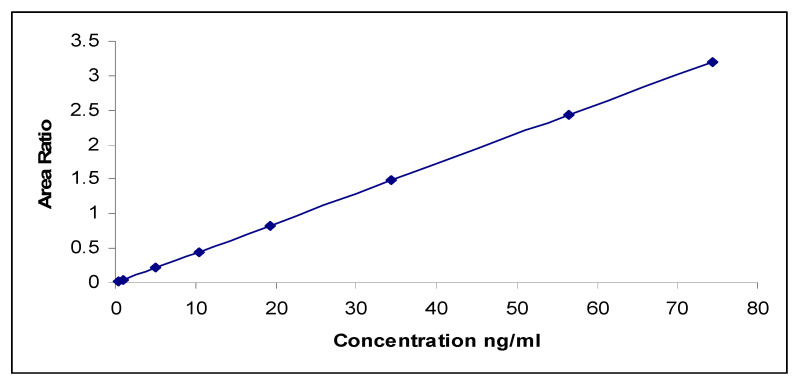
Calibration curve for the mean area ratio of estragole concentration (ranging from 0.50 to 75.0 ng/mL).

**Figure 6 microorganisms-11-01024-f006:**
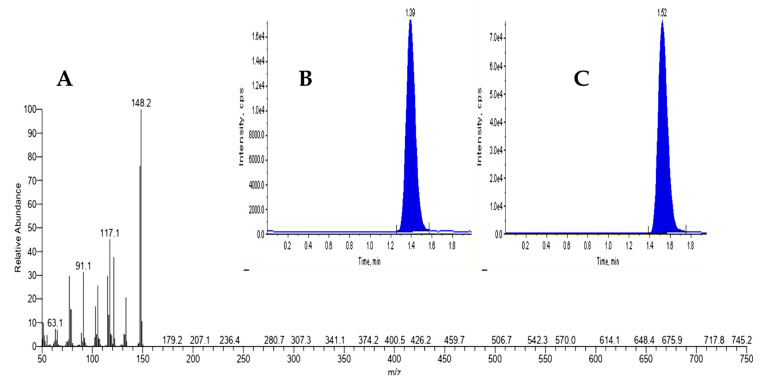
Quantitative analysis of estragole in aniseed. (**A**) Aniseed fragment ions for estragole standard. (**B**) Estragole peak retention time was revealed at 1.39. (**C**) *p*-anisaldehyde internal standard peak retention time was observed at 1.52.

**Figure 7 microorganisms-11-01024-f007:**
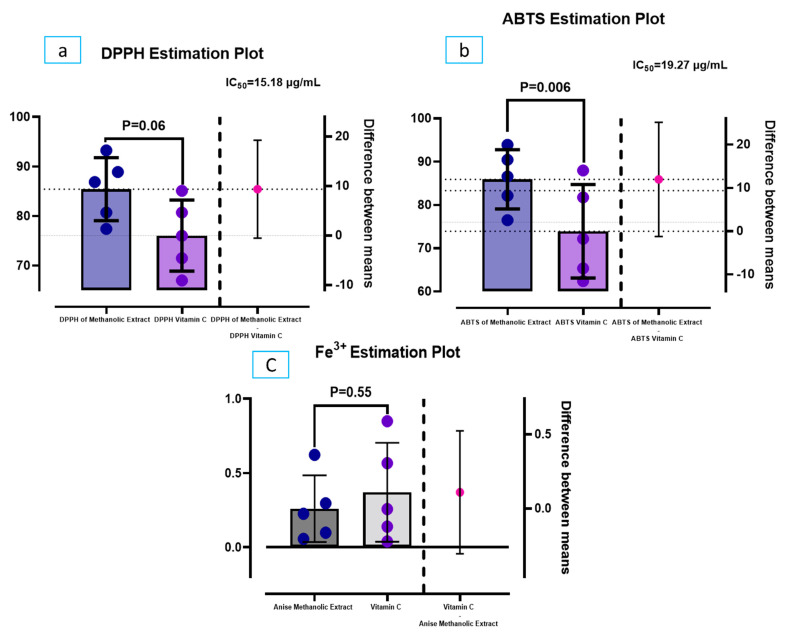
(**a**) DPPH scavenging assay of aniseed methanolic extract and Vit C. (**b**) ABTS scavenging assay of aniseed methanolic extract and Vit C. (**c**) Reducing power scavenging assay of aniseed methanolic extract and Vit C. Results are mean ± SD of five equivalent measurements. Experiment was carried in triplicate. *p* value < 0.05 when compared to Vit C.

**Table 1 microorganisms-11-01024-t001:** The antibiotic susceptibility test for all studied isolates.

Isolate Code	P	AMC	VA	AX	SAM	F	ATM	CPZ	C	DA	CN	TE	E	OFX	NOR	SXT
*E. faecalis* *n* = 4	100	80	100	75	88	91	100	37	78	91	99	91	79	100	100	100
*S. aureus* *n* = 11	100	95	92	94	79	49	I00	74	71	87	97	79	84	99	100	100
*S. typhi* *n* = 9	100	100	78	91	92	69	92	97	93	98	99	72	88	100	92	94
*P. aeruginosa* *n* = 7	100	92	91	89	79	55	90	88	44	94	100	66	91	93	88	93
*A. baumannii* *n* = 6	100	97	49	I	59	79	100	81	57	96	95	49	100	97	100	98

Abbreviations: *E. faecalis*: Enterococcus faecalis; *S. aureus*: Staphylococcus aureus; *S. typhi*: Salmonella typhi; *P. aeruginosa*: Pseudomonas aeruginosa; *A. baumannii*: Acinetobacter baumannii; P: penicillin; AMC: amoxicillin/clavulanate; VA: vancomycin; SAM: ampicillin/sulbactam; F: nitrofurantoin; ATM: aztreonam; CPZ: cefoperazone; C: chloramphenicol; CN: gentamicin; TE: tetracycline; E: erythromycin; OFX: ofloxacin; NOR: norfloxacin; and SXT: trimethoprim/sulfamethoxazole.

**Table 2 microorganisms-11-01024-t002:** VITEK 2 system for the identification and susceptibility testing of *Enterococcus faecalis*.

Antimicrobial	MIC	Interpretation	Antimicrobial	MIC	Interpretation
ESBL	POS	+	Meropenem	1 ≤ 0.25	R
Ampicillin	≥32	R	Amikacin	≥2	R
Ampicillin/sulbactam	4	S	Gentamicin	≥16	R
Clindamycin	≥64	S	Erythromycin	8	R
Ceftriaxone	≥64	R	Ciprofloxacin	≤4	R
Cefepime	2	I	Norfloxacin	≤64	R
Aztreonam	16	R	Vancomycin	≤0.5	R
Ertapenem	≤0.5	R	Nitrofurantoin	≤16	R
Imipenem	≤0.25	R	Trimethoprim/sulfamethoxazole	≤32	R

**Table 3 microorganisms-11-01024-t003:** VITEK 2 system for the identification and susceptibility testing of *Staphylococcus aureus*.

Antimicrobial	MIC	Interpretation	Antimicrobial	MIC	Interpretation
ESBL	POS	+	Meropenem	1 ≤ 0.25	R
Ampicillin	≥32	R	Amikacin	≥2	R
Ampicillin/sulbactam	4	S	Gentamicin	≥16	R
Clindamycin	≥64	R	Erythromycin	8	I
Ceftriaxone	≥64	R	Ciprofloxacin	≤4	R
Cefepime	2	R	Norfloxacin	≤64	R
Aztreonam	16	R	Vancomycin	≤0.5	S
Ertapenem	≤0.5	R	Nitrofurantoin	≤16	R
Imipenem	≤0.25	R	Trimethoprim/sulfamethoxazole	≤32	R

**Table 4 microorganisms-11-01024-t004:** VITEK 2 system for the identification and susceptibility testing of *Salmonella typhi*.

Antimicrobial	MIC	Interpretation	Antimicrobial	MIC	Interpretation
ESBL	POS	+	Meropenem	1 ≤ 0.25	R
Ampicillin	≥32	R	Amikacin	≥2	R
Ampicillin/sulbactam	4	S	Gentamicin	≥16	R
Clindamycin	≥64	R	Erythromycin	8	R
Ceftriaxone	≥64	I	Ciprofloxacin	≤4	R
Cefepime	2	R	Norfloxacin	≤64	R
Aztreonam	16	R	Vancomycin	≤0.5	I
Ertapenem	≤0.5	R	Nitrofurantoin	≤16	R
Imipenem	≤0.25	R	Trimethoprim/sulfamethoxazole	≤32	R

**Table 5 microorganisms-11-01024-t005:** VITEK 2 system for the identification and susceptibility testing of *Pseudomonas aeruginosa*.

Antimicrobial	MIC	Interpretation	Antimicrobial	MIC	Interpretation
ESBL	POS	+	Meropenem	1 ≤ 0.25	R
Ampicillin	≥32	R	Amikacin	≥2	R
Ampicillin/sulbactam	4	R	Gentamicin	≥16	R
Clindamycin	≥64	R	Erythromycin	8	I
Ceftriaxone	≥64	R	Ciprofloxacin	≤4	R
Cefepime	2	R	Norfloxacin	≤64	S
Aztreonam	16	R	Vancomycin	≤0.5	S
Ertapenem	≤0.5	R	Nitrofurantoin	≤16	R
Imipenem	≤0.25	R	Trimethoprim/sulfamethoxazole	≤32	R

**Table 6 microorganisms-11-01024-t006:** VITEK 2 system for the identification and susceptibility testing of *Acinetobacter baumannii*.

Antimicrobial	MIC	Interpretation	Antimicrobial	MIC	Interpretation
ESBL	POS	+	Meropenem	1 ≤ 0.25	R
Ampicillin	≥32	R	Amikacin	≥2	S
Ampicillin/sulbactam	4	R	Gentamicin	≥16	S
Clindamycin	≥64	R	Erythromycin	8	R
Ceftriaxone	≥64	R	Ciprofloxacin	≤4	S
Cefepime	2	R	Norfloxacin	≤64	R
Aztreonam	16	R	Vancomycin	≤0.5	R
Ertapenem	≤0.5	R	Nitrofurantoin	≤16	R
Imipenem	≤0.25	R	Trimethoprim/sulfamethoxazole	≤32	R

**Table 7 microorganisms-11-01024-t007:** Antimicrobial activity of aniseed methanolic extract against multidrug-resistant bacteria.

Isolate	Aniseed Methanolic Extract	Amikacin (10 µg/disc)	Negative Control (DMSO, 100 µL)
*Enterococcus faecalis*	18.32 ± 0.5 ^ab^	14.27 ± 0.2 ^ab^	0.0 ± 0.0
*Staphylococcus aureus*	19.88 ± 0.3 ^ab^	16.23 ± 0.1 ^ab^	0.0 ± 0.0
*Salmonella typhi*	21.33 ± 0.5 ^ab^	20.31 ± 0.3 ^ab^	0.0 ± 0.0
*Pseudomonas aeruginosa*	22.86 ± 0.5 ^ab^	18.56 ± 0.2 ^ab^	0.0 ± 0.0
*Acinetobacter baumannii*	17.56 ± 0.2 ^ab^	15.50 ± 0.5 ^ab^	0.0 ± 0.0

The mean values (within rows) with different superscripts are significantly different (*p* < 0.05).

**Table 8 microorganisms-11-01024-t008:** MIC and MBC of aniseed extract against isolated bacteria.

Isolate	MIC (µg/mL)	MBC (µg/mL)
*Enterococcus faecalis*	95	190
*Staphylococcus aureus*	125	250
*Salmonella typhi*	130	260
*Pseudomonas aeruginosa*	125	250
*Acinetobacter baumannii*	170	340

**Table 9 microorganisms-11-01024-t009:** HPLC/MS analysis of aniseed extract.

Phenolic Compounds	Conc. (µg/g)	RT (min)
*Cis*-anethole	3150.11	2.82
trans-anethole	2312.11	3.55
Ellagic acid	623.54	4.01
Catechin	357.23	4.52
Kaempferol-*O*-rutinoside	96.32	5.24
Syringic acid	521.49	5.92
Pyro catechol	578.31	6.04
Eugenol rutinoside	234.56	7.13
*O*-coumaric acid	578.77	8.11
Vanillin	52.36	8.31
*Trans*-ferulic acid	324.57	9.32
Naringenin	211.23	9.53
Caffeic acid	652.32	9.75
Rosmarinic acid	96.32	10.55
Quercitin rhamnoside	632.32	10.17
Estragole	4422.39	11.52
Quercitin rhamnoside	632.32	12.11
*Trans*-cinnamic	37.25	12.97
Luteolin	23.65	13.92
Apegenin-7-*O*-glucoside	55.62	15.72

**Table 10 microorganisms-11-01024-t010:** Chemical composition of aniseed extract After GC analysis.

Identification	RT (min)	Area %	Peak Area
Linolenic acid	5.38	0.37	10,147,157.35
Eucalyptol	5.45	1.08	29,892,305.40
L-Fenchone	6.70	3.73	103,381,896.04
2-9,12-Octadecadienyloxy	8.11	0.42	11,620,299.63
Estragole	9.49	66.85	1,853,324,123.57
5,7-Dodecadiyn-1,12-diol	13.48	0.36	9,973,126.83
Caryophyllene	15.03	0.95	26,230,030.91
1,3-Benzodioxole	19.76	0.43	11,818,138.28
Gibberellic acid	20.63	.46	12,978,627.00
Pentadecanoic acid	26.29	1.05	29,100,222.44
Palmitic acid	28.60	4.26	118,101,239.86
7,10-octadecadienoic acid	29.42	0.88	24,453,817.51
9-octadecenoic acid	29.58	2.35	65,019,938.64
Cyclopropanebutanoic acid	30.08	0.36	9,960,494.88
9,12-octadecadienoic acid	31.53	0.57	15,750,740.24
Petroselinic acid	31.63	2.72	75,269,544.77
Isochiapin b	32.79	0.24	6,778,401.90
1-Heptatriacotanol	33.00	0.61	17,018,655.01
Hexadecadienoic acid	35.52	0.30	8,369,888.17
Linoleic acid ethyl ester	35.62	0.33	9,209,906.53
cis-13-Eicosenoic acid	36.15	0.38	10,547,993.14
9-octadecenoic acid	36.31	1.17	32,477,736.34
Cyclopropane decanoic acid	36.73	0.72	19,892,392.94
9-hexadecenoic acid	37.11	3.61	100,140,171.41
9-Hexadecenoic acid, 9-octadecenyl ester	38.55	0.96	26,601,291.25
Oleyl oleate	39.14	1.94	53,776,181.23
Nonadecatriene-5,14-diol	39.68	2.90	80,466,096.09

**Table 12 microorganisms-11-01024-t012:** Fragmentation ion and the retention time for estragole standard as well as the quantitative assessment of estragole in aniseed.

Aniseed Retention Time and Fragment Ions for Estragole Standard			
Compound	M.wt	(*m*/*z*)	RT (Min)
Estragole	148.2	150	1.39
**Quantification Results of Estragole in Aniseed**			
**Compound**	**Linear Range (ng/mL)**	**R^2^**	**Recovery (%)**
Estragole	0.5–75	0.9992	101.78% to 104.66%

Abbreviations: M.wt: molecular weight; RT: retention time; R^2^: correlation coefficient.

## Data Availability

All data generated or analyzed during this study are included in this published article (and its Appendix A).

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
