# Peer review of "Chemical Constituents, Antioxidant Potential, and Antimicrobial Efficacy of Pimpinella anisum Extracts against Multidrug-Resistant Bacteria"

_microorganisms, 2023, doi:10.3390/microorganisms11041024_

Round 1
Reviewer 1 Report (Previous Reviewer 3)
The manuscript was corrected in a few points, however, still, the most important question was not corrected.
Quantitative analysis is still not described. It is important information, especially when extracts were analyzed. The matrix effect is also important.
I can agreed that the extract has antibacterial activity, but the question is how it is connected with HPLC and GC data. All interpretation looks like taken from literature. There is no conclusion of the Authors about the connection between analyzed compounds and antibacterial activity from this study. It is the confirmation of literature data [50-70]
Author Response
I upload the reply as attach

Reviewer 2 Report (New Reviewer)
Reviewer comments
Manuscript: microorganisms-23043194 - Chemical Constituents, Antioxidant Potential, and Antimicrobial Efficacy of Pimpinella Anisum Extracts Against Multidrug-Resistant Bacteria.
The authors studied the effectiveness of aniseed methanolic extract against multidrug-resistant bacteria and the antioxidant potential of the extract and its phytochemical composition by using comprehensive GC-MS and HPLC methods. Using HPLC, oxygenated monoterpenes represented the majority of the aniseed content, mainly estragole, cis-anethole, and trans-anethole at 4422.39, 3150.11, and 2312.11 (g/g), respectively. All of the examined bacteria are very sensitive to aniseed's antibacterial effects. Aniseed's antibacterial activity could be attributed to the presence of phenolic compounds which include catechins, methyl gallates, caffeic acid and syringic acids. According to the GC analysis, several flavonoids were detected, including catechin, isochiapin, and trans ferulic acid, as well as quercitin rhamnose, kaempferol-O-rutinoside, gibberellic acid, and hexadecadienoic acid. Utilizing three methods, the extract demonstrated strong antioxidant activity. Aniseed extract clearly inhibited MDR bacterial isolates, indicating its potential use as an anti-virulence strategy. It is assumed that polyphenolic acids and flavonoids are responsible for this activity. Trans-anethole and estragole were aniseed chemotypes. Aniseed extracts showed higher antioxidant activity than vitamin C.
The data analysis methods are correct. There is control. The antibiotic Amikacin was utilised as a reference antibiotic, while an extract-free solution served as the negative control.
The English of the text is well written and well readable but needs additional checking with a professional translator.
The uniqueness of the text is more than 90% by AntiPlagiarism.NET.
The text contains some misspellings and typos. Also need to expand the part of the discussion.
There are some comments and questions:
1) Please explain, why the authors used methanolic extract of aniseed?
2) Why the ability to scavenge free radicals of aniseed was compared to ascorbic acid?
3) What is novelty of this manuscript. Everything about aniseed is known now.
4) Identification of bacterial isolates by VITEK2 system Suspensions. Why the authors did not identify species and lines of bacteria by sequencing DNA?
5) Why the authors did not check the presence of antibiotic resistance genes in bacteria?
6) The aniseed plants grew in different places can differ in their containing compounds biochemical composition and concentration, it depends from climatic condition. Thus any aniseeds cannot be used in medicine for antibacterial treatment without Gas chromatography-Mass Spectroscopy evaluation a biochemical composition.
7) Lines 438-440 - After the sentence - According to this study [70], aniseed polar fractions may be useful as antibiotics and antioxidants in biopharmaceuticals. - add sentence - As alternative to the plant extracts, the natural metabolites of some lactobacilli possess antimicrobial, antifungal, antioxidant and antiinflammatory effects and can be used as metabiotics in medicine (Danilenko et al., 2021)
8) Add to the References: Danilenko, V.N.; Devyatkin, A.V.; Marsova, M.V.; Shibilova, M.U.; Ilyasov, R.A.; Shmyrev, V.I. Common inflammatory mechanisms in COVID-19 and Parkinson’s diseases: the role of microbiome, pharmabiotics and postbiotics in their prevention. J Inflamm Res 2021, 14, 6349–6381, doi:10.2147/JIR.S333887.
Please improve the manuscript according to the above comments and answer questions.
Author Response
I uploaded the file as attach

Round 2
Reviewer 1 Report (Previous Reviewer 3)
The manuscript was successfully corrected. The Author's response is informative. I can recommend the manuscript for publication.
This manuscript is a resubmission of an earlier submission. The following is a list of the peer review reports and author responses from that submission.
Round 1
Reviewer 1 Report
Dear colleagues,
I thank you for accepting the comment. You did a great job of making the corrections and changing the wrong statements.
All the best in your future work
Reviewer 2 Report
This study is very weak and more suitable to lower journal. The authors just characterized the anism extract and tested against some MDR strains. The exact component of achieving activity is still unclear. The identification of the isolated strains is not fully documented and must be identified using 16S rRNA and the exact sequence for identification should be submitted to gene bank.
The tables of Vitek identification Tables 2-5 could be deleted and the repetition of experiment should be performed for more confirmation. Comparison between these data and previously published data should be clarified to indicate the novelty (if any) of this work. The discussion and the abstract are very poor and not informative of the obtained results or impact of the work.
Reviewer 3 Report
Manuscript deals with application of Pimpinella Anisum Extracts against bacteria.
Authers applied some methods of investigation and combined some group of the results. However, in my opinion, the result are not proven.
I can agreed that the extract has antibacterial activity, but I am to sutre that it is somehow connected with HPLC, GC or DPPH results.
Detailed comments:
The description of quantitative analysis should be provided.
There is no information about retention in HPLC. Some chromatogram may illustrate the separation.
Line 89-91. Extract was evaporated. After that it is written that solution was stored... It is not clear.
HPLC column have to be corrected.
Line 139 What means aqueous solvent?